# Catalytic Methanation over Natural Clay-Supported Nickel Catalysts

**DOI:** 10.3390/molecules30102110

**Published:** 2025-05-09

**Authors:** Alejandra Cue Gonzalez, Elsa Weiss-Hortala, Quoc Nghi Pham, Doan Pham Minh

**Affiliations:** 1Centre RAPSODEE, IMT Mines Albi, UMR CNRS 5302, Université de Toulouse, Campus Jarlard, CEDEX 09, 81013 Albi, France; 2ICMMO (Institute of Molecular Chemistry and Materials), Université Paris-Saclay, CNRS, 91400 Orsay, France; 3Sustainable Environment Research Institute, Chulalongkorn University, Bangkok 10330, Thailand

**Keywords:** nickel, natural clay, catalysis, catalytic methanation, methane

## Abstract

The catalytic methanation reaction allows for the attainment of methane from carbon dioxide and hydrogen. This reaction is particularly interesting for the direct upgrading of biogas, which mainly contains methane and carbon dioxide, into biomethane. This work focused on the synthesis and evaluation of natural clay-supported nickel catalysts in the catalytic methanation reaction. Natural clay could be directly used as a low-cost catalyst support for the deposition of small nickel nanoparticles (1–15 nm) by the standard incipient wetness impregnation method. These catalysts showed high activity and excellent selectivity into methane and excellent catalytic stability (80% carbon dioxide conversion, nearly 100% methane selectivity at 500 °C, 1 bar, and WHSV = 17,940 mL·g_cat_^−1^·h^−1^ for 48 h on stream) and outperformed their counterparts prepared with an industrial alumina support as reference.

## 1. Introduction

Biogas is the main product of the transformation of biomass and biowaste by anaerobic digestion [1]. According to the International Energy Agency (IEA), worldwide biogas production has largely increased during the recent decades [2]. Raw biogas not only contains methane (ca. 50–70 vol.%) but also carbon dioxide (ca. 30–50%) and other undesirable molecules (e.g., CO_2_, H_2_S, NH_3_, H_2_O, siloxanes, volatile organic compounds, etc.) [3,4]. To date, one of the main applications of biogas is to produce biomethane for injection into the gas grid. To that end, undesirable molecules including large portions of carbon dioxide are removed to meet the quality standards required for gas grid injection. As an example, there are currently more than 600 sites of biomethane injection into the gas grid in France [5].

According to the second principle of green chemistry, it is meaningful to valorize the carbon dioxide in raw biogas into valuable products instead of separating and releasing it into the atmosphere. The direct hydrogenation of carbon dioxide into methane (methanation, Equation (1)) seems to be the most relevant process, since the main product is also methane. This allows for the production of more renewable methane when hydrogen used in the methanation process is also produced from a renewable energy source such as solar energy, wind energy, geothermal energy, etc. Also, it results in reductions in carbon emissions.CO_2_ + 4H_2_ → CH_4_ + 2H_2_O  ΔH° = −165 kJ/mol(1)

Due to its slow kinetics, the methanation reaction requires suitable catalysts to reach reasonable reaction rates under thermodynamically favored conditions [6]. However, the most efficient catalysts involve costly noble metals, e.g., Ru- and Rh-based catalysts [7,8]. This may hinder the development of large-scale projects. Moreover, these metals have high potential environmental impacts, which is not in line with the international objectives set to provide access to clean and affordable energy [9]. Therefore, transition metals such as Ni that are more accessible appear to be the best candidates for the design of a competitive methanation catalyst [10,11,12]. Indeed, nickel-based catalysts have been largely studied in catalytic methanation [13]. Usually, nickel is dispersed as nanoparticles at the surface of a catalyst support. The catalyst support fulfills the role of enhancing the catalyst activity by dispersing the metal (Ni) species, influencing their morphology, improving their stability, modifying the reducibility of metal oxide precursors, promoting metal–support interactions, favoring electron transfer between the metal and the support, etc. [12,14,15,16,17]. These supports must not only enhance the catalytic performance of the catalyst but also be accessible in terms of costs and resource availability.

Traditional supports such as alumina are industrially available, enabling high CO_2_ conversion and CH_4_ selectivity performance when paired with an active nickel (Ni) phase. However, alumina is industrially produced by processes with high environmental impacts such as the Bayer process [18,19]. Furthermore, the synthesis of Ni catalysts supported on alumina typically involves a nickel nitrate precursor, which requires high-temperature calcination (up to 900 °C) to form the active catalytic phase [20]. This step is often the most environmentally intensive part of the catalyst production process, with substantial greenhouse gas emissions [13,20]. In accordance with the third principle of green chemistry, which emphasizes minimizing harm caused by materials and substances used in synthetic methods, it is essential to explore alternative catalyst support materials. These alternatives should ideally have reduced environmental impacts and lower costs, ultimately enabling the production of more accessible and sustainable catalysts for industrial applications.

Among the possible support materials with these characteristics, natural clays have emerged as an efficient alternative. Natural minerals and clays have demonstrated good catalytic performance in CO and CO_2_ methanation, as explored in a review by Medina and collaborators [13], with studies boasting CO_2_ conversion rates of up to 85 mol% and CH_4_ selectivity of 100 mol% [21,22,23,24]. This is likely due to the inherent characteristics of clay, boasting a possibly high surface area, a mesoporous nature, and a basic surface, which enhance the reducibility and the dispersion of metal particles when used as catalyst supports [13]. Additionally, clays often contain alkaline earth metals like calcium and magnesium, which can create basic sites that facilitate the formation of reactive carbonate species that serve as CH_4_ precursors [21].

Given the potential of clay-based catalysts, our present work focuses on designing a high-performance catalyst for the direct hydrogenation of biogas mixtures composed primarily of CH_4_ and CO_2_. Specifically, we developed a Ni nanoparticle-based active phase supported on natural clay sampled from the South of France. By exploring this alternative support material, we aim to contribute to the development of more sustainable and accessible catalysts for industrial applications.

## 2. Results and Discussion

To assess the catalytic performances of the developed catalysts on the CO_2_ methanation reaction, they are first characterized.

### 2.1. Catalyst Characterizations

Prior to preparing catalysts with Ni deposition, the clay support was characterized. The results of the elemental analysis by XRF are displayed in Table 1 for the natural clay used as catalyst support in this work. As expected, the most prominent elements are Si, Fe, Al, K, and Ca. Silicon has a far greater proportion than the rest of the elements, indicating the presence of silica- and/or aluminosilicate-based components, which are efficient as catalyst support in the methanation reaction [25,26]. Fe, Na, K, Ba, Mg, and Ca are helpful minerals expected to promote catalytic methanation, since they can catalyze this reaction [27,28,29,30,31]. Furthermore, the nickel content was measured by the ICP-AES technique for the calcined catalysts, reaching 8.4 and 5.8 wt.% for Ni/Clay_Cal and Ni/Al_2_O_3__Cal, respectively. In comparison with the theoretically targeted nickel content of 10 wt.%, important loss took place during the impregnation steps, which might have been due to the small size of the prepared samples.

Nitrogen adsorption–desorption isotherms of the initial clay and alumina supports (Appendix A) show two different profiles. The natural clay displays an H3-type hysteresis loop without a saturation plateau at high relative pressure, meaning the pores are not completely saturated by the condensed gas during the analysis [32]. This could indicate that the pores may be slit-shaped, which is characteristic for an unstable structure that expands by capillary condensation. The adsorbing structure forms aggregates with the adsorbate, or the pore size distribution is wide (mesoporous and almost macroporous). The alumina support displays a type IV isotherm, indicating the presence of mesopores in this material. Also, this support has an H1-type hysteresis loop with a saturation plateau at a high relative pressure and almost parallel adsorption and desorption curves. This suggests the presence of narrow and potentially cylindrical mesopores [32]. According to these nitrogen adsorption–desorption isotherms, the specific surface area and the total porous volume of both the supports reached 130 m^2^/g and 0.24 cm^3^/g for alumina and 20 m^2^/g and 0.16 cm^3^/g for the natural clay, respectively. In terms of surface texture, alumina would be a more suitable support for Ni impregnation.

SEM images of the initial supports and the dried and calcined catalysts are presented in Figure 1. The natural clay is mostly composed of sheet-like particles (Figure 1a), confirming the discussion of the nitrogen adsorption–desorption isotherm (Appendix A). The surface of the dried Ni/Clay catalyst (Figure 1b) is likely similar to that of the initial clay support, with random dispersion of nickel precursor deposits with filamentous shapes throughout the surface. The sheet-like structure of the initial support is still visible below these crystallites. For the calcined Ni/Clay_Cal catalyst (Figure 1c), nickel oxide particles, formed by the thermal decomposition of the nickel precursor at around 400 °C [33] (highlighted by TG analysis in Appendix A), are spread throughout the surface. With respect to alumina-based materials, it is difficult to observe the microstructure of the initial alumina via SEM image (Figure 1d). However, deposits of the nickel precursor can be observed on the surface of the dried Ni/Al_2_O_3_ catalyst (Figure 1e). Finally, small nickel oxide particles, formed by the thermal decomposition of the nickel precursor at around 400 °C [33,34] (Appendix A), are regularly distributed on the surface of the calcined Ni/Al_2_O_3__Cal catalyst. For both the calcined catalysts, the size of nickel oxide particles seems to be roughly dozens of nm, and the Ni/Al_2_O_3__Cal catalyst seems to have smaller nickel oxide nanoparticles than the Ni/Clay_Cal catalyst.

Figure 2 shows TEM images of the freshly reduced Ni/Clay (Figure 2a,b) and Ni/Al_2_O_3_ (Figure 2d,e) catalysts, as well as the nickel particle-size distributions (Figure 2c,f). Metallic particles, as indicated by red arrows in Figure 2, of 1–9 nm are the most abundant, but larger particles are also present—in particular, for the case of the freshly reduced Ni/Clay catalyst. Large nickel particles might be due to the elongated shapes of the clay support, which favor the formation of these particles. The latter could be formed in the pores and interparticle spaces created by the sheet-like structures present in the clay support material, as observed in the SEM analysis. The nickel particle-size distributions in Figure 2 show that, Ni/Al_2_O_3__Cal contains smaller nickel nanoparticles than Ni/Clay_Cal, as previously observed by SEM (Figure 1). This could be explained by a high specific surface area of the alumina support (120 m^2^ g^−1^) in comparison to clay (20 m^2^ g^−1^).

Figure 3a shows the TPR profiles of the dried catalysts. On the alumina-supported nickel catalyst, only a reduction peak, centered at 317 °C, is observed, which must correspond to the reduction of Ni^2+^ species into metallic Ni. In the case of the clay-supported nickel catalyst, two reduction peaks, centered at 358 and 386 °C, are observed, which suggest either interactions of Ni^2+^ species with different superficial species of the clay support or the stepwise reduction of Ni^2+^ species on the surface of the clay support. Moreover, the higher reduction temperatures observed for the clay-supported nickel catalyst suggest stronger interactions of nickel species with the clay support than with the alumina support. In Figure 3b, the reduced Ni/Clay catalyst shows a large CO_2_ desorption peak centered at 590 °C and possibly another small CO_2_ desorption peak at 755 °C. In the case of the reduced Ni/Al_2_O_3_ catalyst, a small CO_2_ desorption peak is observed at a low temperature of 121 °C, followed by a large peak within the range of 165–400 °C and an intense peak centered at 570 °C. Globally, the desorption of CO_2_ took place at a higher temperature over the Ni/Clay catalyst versus the Ni/Al_2_O_3_ catalyst, indicating a high basicity strength of the Ni/Clay catalyst.

The XRD patterns of the initial supports and the catalysts before and after air calcination are presented in Figure 4. The natural clay (Figure 4a) mostly contains quartz (SiO_2_) as the main crystalline phase, followed by muscovite, a hydroxylated potassium aluminum silicate [(KF)_2_(Al_2_O_3_)_3_(SiO_2_)_6_(H_2_O)]. Another potential component is albite (NaAlSi_3_O_8_), whose main peaks at 2θ = 22.1 and 27.9 degrees are superposed or close to those of quartz and muscovite. Some other small peaks could not be identified, which might be due to the presence of other minor crystalline phases in the natural clay. The dried catalyst (Ni/Clay) shows an XRD pattern similar to that of the natural clay. For the calcined catalyst (Ni/Clay_Cal), the presence of NiO crystallites was also confirmed via its characteristic peaks at 2θ = 37.2, 43.3, and 62.8 degrees (reference JCPDS 00-044-1159).

As shown in Figure 4b, the initial alumina was still in the form of alumina hydrate (boehmite, main peaks at 2θ = 14.5, 28.2, 38.4, and 49.3 degrees), which is consistent with the information from the provider (Sasol). The dried catalyst (Ni/Al_2_O_3_) shows peaks similar to those of the initial support but with some small peaks from the nickel precursor (Ni(NO_3_)_2_·(H_2_O)_4_, with the main peaks at 2θ = 18.2, 20.16, 22.5, 26.2, 30.7, and 34.8 degrees). On the other hand, the calcined catalyst (Ni/Al_2_O_3__Cal) mostly contains a γ-alumina crystalline phase (γ-Al_2_O_3_, with main peaks at 2θ = 19.3, 31.8, 37.3, 45.3, 60, and 66.6 degrees), showing the thermal transformation of this support to eliminate all the physiosorbed and crystal water, as confirmed by TG analysis (Appendix A). As expected, the crystalline phase of nickel oxide was well formed in this calcined catalyst, confirming the SEM results presented in Figure 1. In addition, the broad NiO peaks denote the formation of fine crystallites, which likely results in fine particles, confirming TEM results on the formation of small nickel nanoparticles (see Figure 2).

As mentioned in Section 3.3, prior to the catalytic tests, the dried or calcined catalysts were reduced in situ under hydrogen flux. Thus, XRD analysis of the reduced catalysts was also performed in order to confirm the formation of metallic nickel nanoparticles. All four catalysts (Ni/Clay, Ni/Clay_Cal, Ni/Al_2_O_3_, and Ni/Al_2_O_3__Cal) were reduced under 5 vol.%H_2_/Ar at 500 °C for 2 h and cooled down under the same gas mixture. At room temperature, XRD patterns of the reduced catalysts were quickly recorded. Appendix A compares the raw XRD results of each catalyst before and after H_2_ reduction. As expected, peaks of metallic nickel were systematically observed for the reduced catalysts. According to the main diffraction peaks of the metallic nickel ((111) at 44.5°, (200) at 51.8°, and (220) at 76.4°; reference JCPDS 00-004-0850), the size of Ni crystallites could be determined by the Scherrer equation [35], reaching ca. 14, 15.5, 7, and 8 nm for the freshly reduced Ni/Clay, Ni/Clay_Cal, Ni/Al_2_O_3_, and Ni/Al_2_O_3__Cal, respectively. Therefore, the calcination treatment slightly increased the size of nickel crystallites in comparison with the non-calcined counterparts. These results also enhanced the SEM and TEM observations reported in Figure 1 and Figure 2, wherein alumina-based catalysts showed smaller nickel particles than clay-based catalysts.

XPS measurements were performed on the reduced Ni-based catalysts to investigate the surface chemical states of nickel and understand the mechanisms of those catalysts. The analysis was focused on the Ni 2p region, known for its complex structure involving multiplet splitting, shake-up satellites, and charge effects [36,37]. The fitting procedure was carried out using reference Shirley line shapes extracted from standard compounds of metallic Ni, NiO, and Ni(OH)_2_ measured on the same spectrometer under identical conditions. These references, which are considered highly reliable, were used to guide the deconvolution of the experimental spectra (see Appendix A).

Figure 5 shows the deconvolution of the Ni 2p_3/2_ spectra, which revealed the presence of three main species in both samples: metallic Ni (~852.4–852.6 eV), NiO (~854.4 eV), and Ni(OH)_2_ (~856.0 eV). These components were evidently dominant and sufficient to describe the spectra. For the Ni/Clay sample, the surface Ni⁰ fraction accounted for approximately 20% of total Ni, while in Ni/Al_2_O_3_, it was lower, at around 11%. This difference is consistent with the XRD-derived crystallite sizes reported above: although Ni particles on Al_2_O_3_ are smaller (7–8 nm) than on clay (14–15 nm), XPS only probes the outer ~3–5 nm of each particle. Thus, a 1–2 nm oxide/hydroxide shell consumes most of the XPS-visible volume on the small Al_2_O_3_-supported Ni, leaving a small metallic Ni^0^ signal (~11%). On the larger clay-supported Ni, the same shell is a thinner “skin” relative to the particle size, so more Ni⁰ remains within the XPS sampling depth, resulting in a higher Ni^0^/Ni_total ratio (~20%). Additionally, stronger Ni–Al_2_O_3_ interactions can form irreducible NiAl_2_O_4_, further reducing the Ni⁰ fraction. In fact, although the spectra were sufficiently fit using only Ni^0^, NiO, and Ni(OH)_2_ components, the potential presence of a NiAl_2_O_4_ spinel phase, especially in the Ni/Al_2_O_3_ sample, cannot be ruled out. This species is often associated with strong metal–support interactions and partial irreducibility, as reported by Li et al. [38]. However, its signature overlaps with those of other Ni^2+^ species and, thus, cannot be unambiguously identified by XPS alone.

### 2.2. Catalytic Performance

Figure 6 show the catalytic behavior of the different catalysts with and without calcination pretreatment. It is worth recalling that all the dried and calcined catalytic materials were reduced in situ under H_2_/N_2_ before catalytic tests. As shown in Figure 6a, when the reaction temperature was fixed at 300 °C, low CO_2_ conversions (3–26%) were obtained. The two dried catalysts (Ni/Clay and Ni/Al_2_O_3_) show higher CO_2_ conversions than the two calcined counterparts (Ni/Clay_Cal and Ni/Al_2_O_3__Cal). The in situ reduction of the dried catalysts might have led to the formation of smaller nickel nanoparticles in comparison to those of the calcined counterparts, but this should be further confirmed by TEM analysis, since TEM analysis was not performed for the dried catalysts. Moreover, the calcination step might also favor the partial formation of solid solutions such as NiAl_2_O_4_, which are much more difficult to reduce [39]. At the investigated reaction temperature (300 °C), the dried Ni/Al_2_O_3_ catalyst was more active than the corresponding dried Ni/Clay catalyst, despite the lower nickel content in Ni/Al_2_O_3_ than in Ni/Clay according to the ICP-AES results. This could be due to the smaller nickel particle size in the Ni/Al_2_O_3_ catalyst compared to that in Ni/Clay and to the fact that the reaction temperature of 300 °C was not enough to activate the various mineral elements found in the clay support (Fe, Na, K, Ba, Mg, and Ca; Table 1). In contrast to the dried catalysts, Ni/Clay_Cal was slightly more active than Ni/Al_2_O_3__Cal, which might be due to the partial formation of a NiAl_2_O_4_ solid solution in the case of the Ni/Al_2_O_3__Cal catalyst during its calcination. In order to better highlight the impact of the clay support on the catalyst’s performance, further catalytic tests at higher reaction temperatures are needed. With respect to the products of the reaction, methane was mainly formed (Figure 6b), and only traces of carbon monoxide could be observed as a byproduct.

The two catalysts without calcination pretreatment (Ni/Clay and Ni/Al_2_O_3_) were further investigated at higher reaction temperatures (400–500 °C), and the results are displayed in Figure 7. The reaction conditions, including the height of the catalyst bed (or the contact time), were similar for each reaction temperature. This allowed for equal comparison of these two catalysts. Thus, in all cases, Ni/Clay was systematically found to be more active than Ni/Al_2_O_3_. Contrary to the tests at 300 °C, mineral impurities such as Na, K, Ba, Mg, Ca, Fe, etc., in the natural clay (Table 1) likely contributed to the methanation reaction as active phases and/or as catalyst promoters [27,28,29,30,31], which explains the superior catalytic activity of the Ni/Clay catalyst. As an example, it was previously experimentally demonstrated that doping a Ni/ZSM-5 catalyst with NaOH led to better catalytic activity in comparison to the counterpart without NaOH treatment in the methanation reaction within the range of 200 to 450 °C and 1 atm [40]. In our case, several SEM-EDX analyses were performed on the surface of the Ni/Clay catalyst, and the results displayed in Appendix A clearly highlight the presence of these minerals on the surface of this catalyst, favoring its methanation activity. The superiority of the catalytic activity of the Ni/Clay catalyst may also be due to its higher total nickel content, as well as its higher superficial metallic nickel content, in comparison with those of Ni/Al_2_O_3_, as revealed by ICP-AES and XPS.

Methane was the main product of the reaction, but small amounts of CO (<1%) were also observed with Ni/Al_2_O_3_, which was not the case for Ni/Clay.

To confirm the superior catalytic activity of Ni/Clay compared to Ni/Al_2_O_3_, two other experiments were performed with a shorter contact time by reducing the catalyst mass by half (500 mg instead of 1000 mg). The results are shown in Figure 8a,b. Once again, Ni/Clay was more active than Ni/Al_2_O_3_. These results confirm the interest in using natural clay as low-cost and highly available catalyst support that contains useful components for the studied catalytic reaction.

Finally, to check the catalytic stability of the Ni/Clay catalyst, another test was performed for 48 h on stream (Figure 8c,d). This test was conducted in the presence of methane (22.6 vol.%, Mixture 2) in the feed. In fact, the composition of Mixture 2 simulates a typical cleaned biogas, mainly containing methane and carbon dioxide. Despite the presence of methane in the feed (which is also the main product of the methanation reaction), carbon dioxide conversion reached 76% at the beginning of the reaction and 72% after 48 h on stream, which is close to the carbon conversion observed in Figure 8a without methane in the feed. TG analysis of the used catalyst recovered after the reaction evidenced that solid carbon deposition was negligible (Appendix A). Moreover, a very good carbon balance was also confirmed for this long test (Appendix A). These results are encouraging for the design of a direct catalytic methanation process for the valorization of cleaned biogas without a CO_2_ separation step.

## 3. Materials and Methods

### 3.1. Catalyst Preparation

A natural clay collected in the south of France was used as the catalyst support. It was dried at 105 °C overnight, then ground and sieved to a maximum particle size of 500 µm. Hereafter, this fraction is called *Clay* and was used as catalyst support for nickel deposition. For comparison, a commercial alumina support (PURAL NW from Sasol) was used as a reference, hereafter referred to as Al_2_O_3_.

The catalysts were prepared using the standard incipient wetness impregnation method (IWI) using nickel(II) nitrate hexahydrate (Ni(NO_3_)_2_·6H_2_O) as a nickel precursor, targeting 10 wt.% nickel loading for each catalyst. This method was chosen because it is largely used for the preparation of supported catalysts. After the impregnation step, the mixtures were dried overnight at 105 °C. Two dried catalysts were obtained, named Ni/Clay and Ni/Al_2_O_3_. Half of each dried catalyst was calcined in a muffle furnace under static air at 500 °C for 3 h and freely cooled down to room temperature. These calcined catalysts were named Ni/Clay_Cal and Ni/Al_2_O_3__Cal.

### 3.2. Characterization Analysis Techniques

Inductively Coupled Plasma–Atomic Emission Spectroscopy (ICP-AES) was performed on a HORIBA Jobin Yvon Ultima 2 (Paris, France) apparatus to determine the nickel contents of the catalysts. Nickel nanoparticles on the catalyst surface were completely dissolved in concentrated inorganic acid mixtures before ICP-AES analysis. The specific surface area and pore-size distribution of the samples were determined using nitrogen adsorption–desorption isotherms with the Brunauer–Emmett–Teller (BET) method. The analysis was performed on a BET Tristar II 3020 from Micromeritics (Norcross, GA, USA). Prior to analysis, the samples were outgassed at 105 °C for 30 h to remove any impurities. Nitrogen adsorption was then carried out at 77 K. Thermogravimetric analysis coupled with differential scanning calorimetry (TG-DSC) was performed using TGA-DSC 111 from Setaram (Caluire et Cuire, France) (from 30 to 800 °C at 5 °C/min). Scanning electron microscopy (SEM) was used to observe the morphology and surface texture of the samples with an Environmental SEM Quattro ESEM apparatus from Thermo Fischer Scientific (Strasbourg, France). X-ray diffraction (XRD) was used to identify the crystalline phases in the samples. The XRD patterns were recorded on an X’PERT PRO MDP from Philips PANalytical in the 2θ range of 6.5–90 degrees with an increment of 0.033 degrees and an acquisition time of 219.7 s per step. A Cu tube was used as the X-ray source, operating at an intensity of 40 mA and a tension of 45 kV. During XRD measurements of Ni/support samples reduced under Ar/H_2_ flow at a high temperature, specific precautions were taken to minimize the rapid oxidation of metallic Ni upon exposure to air. Therefore, the diffraction patterns were recorded in the 2θ range of 10–80°, with a step size of 0.039° and a counting time of 240 s per step. X-ray fluorescence spectroscopy (XRF) was employed to determine the elemental composition of the catalysts using an Epsilon 3XLE benchtop energy-dispersive X-ray fluorescence (EDXRF) spectrometer. Temperature-programmed reduction (TPR) of the dried catalysts was carried out with an AutoChem II/HP from Micromeritics, in which the dried catalysts were heated from 50 to 900 °C (10 °C/min heating rate) under 5 vol.%H_2_/N_2_ flow (100 mL/min). Temperature-programmed desorption of CO_2_ was also performed with this same AutoChem II/HP equipment. First, the dried catalysts were reduced in situ at 450 °C for 2 h under 5 vol.%H_2_/N_2_ flow (100 mL/min). Then, the catalysts were cooled down to 50 °C under N_2_ flow. At this temperature, the reduced catalysts were saturated with 5 vol.%CO_2_/N_2_ flow (100 mL/min). After purging with He flow (100 mL/min) for 30 min, CO_2_ desorption took place by increasing the temperature from 50 to 900 °C (10 °C/min heating rate). The XPS spectra were recorded by a Thermo Fisher Scientific (Courtaboeuf, Les Ulis, France) spectrometer equipped with an Al Kα monochromatic high-energy radiation source (hʋ = 1486.7 eV) and a hemispherical analyzer operating in Constant Analyzer Energy (CAE) mode. XPS data were analyzed using CASA XPS software version 2.3.25PR1.0 (Clearwater, FL, USA). Survey scans were conducted with a pass energy of 200 eV and a step size of 1 eV. High-resolution windows were acquired with a pass energy of 50 eV and a step size of 0.1 eV. The XPS spectra were calibrated using the Al 2p peak at 74.7 eV rather than the commonly used adventitious C 1s signal. This approach is more reliable for alumina- or clay-containing samples, as the Al 2p peak is less affected by surface charging and contamination. This choice is well established and has been validated in prior studies [38].

### 3.3. Catalytic Tests

Appendix A illustrates the fixed-bed rector used for the catalytic experiments. The synthetic gaseous reactant mixture was obtained from four gas bottles (CO_2_, H_2_, CH_4_, and N_2_). The individual gas flow rate of each bottle was controlled by a mass flow controller to reach the desired gas composition. Two gas compositions were used. The first one, called Mixture 1 (Table 2), contained 22.0 mL/min CO_2_, 90 mL/min H_2_, 37.5 mL/min N_2_, and 0 mL/min CH_4_ (or 14.7 vol.% CO_2_, 60.2 vol.% H_2_, 25.1 vol.% N_2_, and 0 vol.% CH_4_), and the second one, called Mixture 2 (Table 2) contained 22.0 mL/min CO_2_, 90 mL/min H_2_, 3.75 mL/min N_2_, and 33.75 mL/min CH_4_ (or 14.7 vol.% CO_2_, 60.2 vol.% H_2_, 2.51 vol.% N_2_, and 22.6 vol.% CH_4_), so the total inlet gas flow rate was the same for these two compositions, but Mixture 2 contained methane to assess the impact of its presence in the inlet mixture on the catalytic performance. The gas mixture fed a fixed-bed reactor, which is a tubular quartz tube (8 mm inner diameter and 250 mm length). At the middle of the reactor tube, a porous fritted quartz disc was used to keep the catalyst bed at the center of the reactor. The catalyst bed was composed of a layer of a dried (Ni/Clay or Ni/Al_2_O_3_) or calcined (Ni/Clay_Cal or Ni/Al_2_O_3__Cal) catalyst powder embedded in two layers of inert alumina powder that was previously sintered at 1000 °C for 5 h (BET surface below 3 m^2^/g). For a given methanation test, the reactor tube was heated by an electric furnace (10 °C/min ramp) under a mixture of 20 vol.%H_2_/N_2_ (100 mL/min) up to a desired temperature (400 or 500 °C) and kept at this temperature for 2 h for the in situ reduction step. Then, the reactor temperature was adjusted to the desired reaction temperature (300–500 °C), and the reactor was fed with the reactant mixture. The outlet gas passed through a silica gel trap that retained the water vapor produced during the reaction before reaching the event. During the reaction, samples were periodically collected in a special gas bag (Tedlar) and analyzed by a micro gas chromatograph (µ-GC) from SRA Instruments. Nitrogen was used as an internal standard. Table 2 summarizes the reaction conditions employed for all the methanation tests. Tests #1 to #4 were selected to compare different dried and calcined catalysts at 300 °C, tests #5 to #10 served as comparative tests of two dried catalysts under different conditions of reaction temperature and contact time, and the last test (#11) was performed to check the catalyst’s stability during a long reaction time. Based on the µ-GC analysis results, carbon dioxide conversion, methane production, and carbon balance were calculated according to Equations (2) to (5).(2)$${\mathrm{CO}}_{2}\;\mathrm{conversion}:\;X_{CO2}=\frac{{\dot{Q}}_{CO2}^{in}-{\dot{Q}}_{CO2}^{out}}{{\dot{Q}}_{CO2}^{in}}$$(3)$${\mathrm{CH}}_{4}\;\mathrm{production}:\;Y_{CH4}=\;\frac{{\dot{Q}}_{CH4}^{out}}{{\dot{Q}}_{CO2}^{in}-{\dot{Q}}_{CH4}^{in}}$$(4)$$\mathrm{Inlet}\;\mathrm{carbon}\;\mathrm{flow}\;(\mathrm{mmol}/\min):\;C^{in}={\dot{Q}}_{CO2}^{in}+{\dot{Q}}_{CH4}^{in}$$(5)$$\mathrm{Outlet}\;\mathrm{carbon}\;\mathrm{flow}\;(\mathrm{mmol}/\min):\;C^{out}={\dot{Q}}_{CO2}^{out}+{\dot{Q}}_{CH4}^{out}+{\dot{Q}}_{CO}^{out}$$
where ${\dot{Q}}_{CO2}^{in}$ and ${\dot{Q}}_{CO2}^{\mathrm{out}}$ (mmol/min) are the input and output flowrates of carbon dioxide, respectively; ${\dot{Q}}_{CH4}^{\mathrm{in}}$ and ${\dot{Q}}_{CH4}^{\mathrm{out}}$ (mmol/min) are the input and output flowrates of methane, respectively; and ${\dot{Q}}_{CO}^{\mathrm{out}}$ (mmol/min) are the input and output flowrate carbon monoxide.

## 4. Conclusions

A series of nickel-based catalysts was synthesized via impregnation using nickel nitrate as a precursor and two catalyst supports: natural clay and Al_2_O_3_ as reference. The catalytic materials were characterized and evaluated in the CO_2_ methanation reaction. Notably, the results indicate that calcination prior to in situ reduction had a detrimental effect on the catalytic performance. In contrast, direct reduction of the dried catalyst under hydrogen was found to be beneficial. Among the prepared catalysts, non-calcined Ni/Clay emerged as the most effective catalyst under the employed operational conditions, outperforming even the reference catalyst (Ni/Al_2_O_3_). This catalyst exhibited high CO_2_ conversion rates of 80–90% with a 500 °C reduction temperature at a 400 °C reaction temperature and 1 bar pressure, with excellent selectivity towards CH_4_ and only traces of CO. Furthermore, it demonstrated good catalytic stability over an extended period of 48 h on stream for the methanation of a simulated cleaned biogas. This opens new possibilities to design high-performance methanation catalysts, as well as catalysts for other reactions, using low-cost and environmentally friendly natural clay supports.

## Figures and Tables

**Figure 1 molecules-30-02110-f001:**
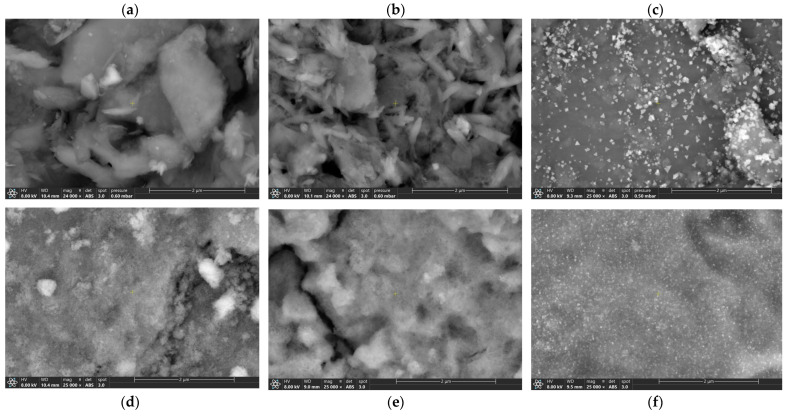
Examples of SEM images of (**a**) dried clay, (**b**) dried Ni/Clay, (**c**) calcined Ni/Clay_Cal, (**d**) alumina, (**e**) dried Ni/Al_2_O_3_, and (**f**) calcined Ni/Al_2_O_3__Cal.

**Figure 2 molecules-30-02110-f002:**
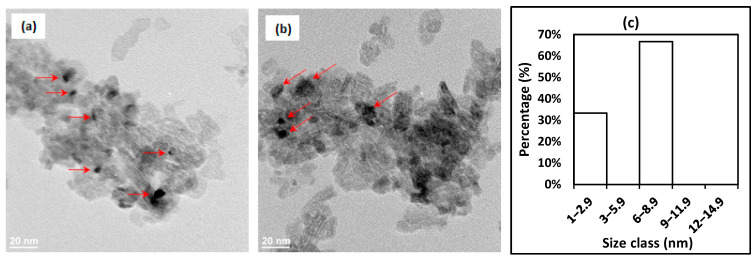
TEM images and Ni particle size distribution of the freshly reduced Ni/Clay catalyst (**a**–**c**) and the freshly-reduced Ni/Al_2_O_3_ catalyst (**d**–**f**). Red arrows indicate the positions of some nickel nanoparticles.

**Figure 3 molecules-30-02110-f003:**
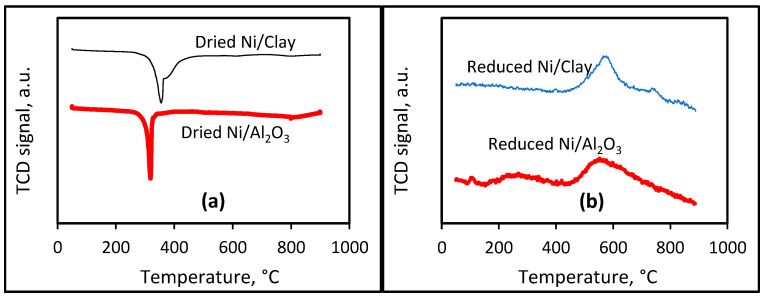
(**a**) TPR profiles of dried catalysts and (**b**) CO_2_-TPD profiles of reduced catalysts.

**Figure 4 molecules-30-02110-f004:**
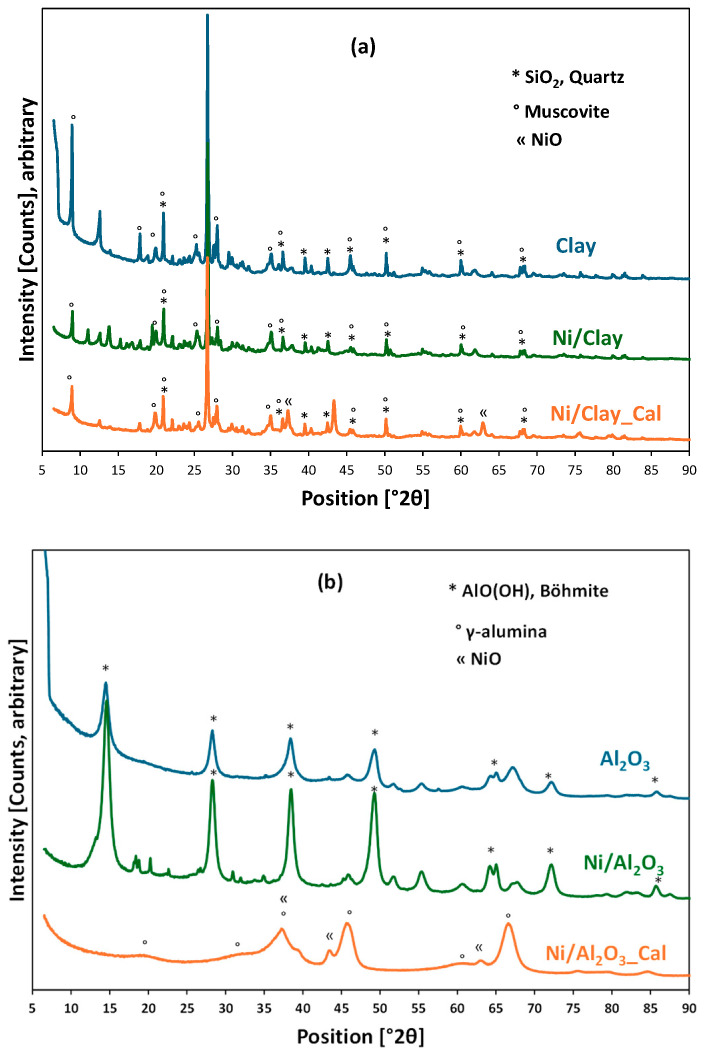
XRD patterns of (**a**) clay-based materials (**b**) and alumina-based materials.

**Figure 5 molecules-30-02110-f005:**
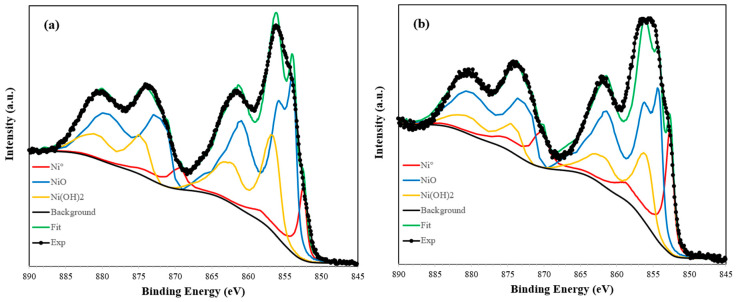
XPS deconvolution of the Ni 2p region for reduced catalysts: (**a**) Ni/Al_2_O_3_ and (**b**) Ni/clay.

**Figure 6 molecules-30-02110-f006:**
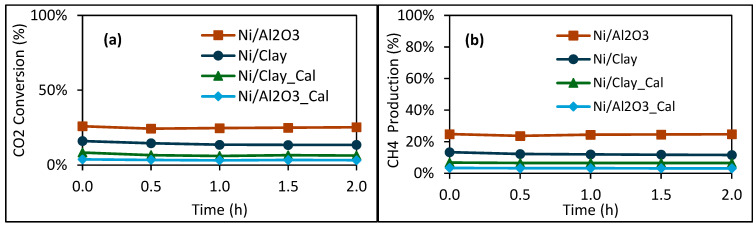
Carbon dioxide conversion (**a**) and methane production (**b**) during catalytic methanation at 300 °C reaction temperature. Other conditions: in situ reduction at 500 °C; 1 g of catalyst; inlet gas flowrate: 22.0 mL/min CO_2_, 90 mL/min H_2_, 37.5 mL/min N_2_, and 0 mL/min CH_4_; WHSV = 8970 mL·g_cat_^−1^·h^−1^.

**Figure 7 molecules-30-02110-f007:**
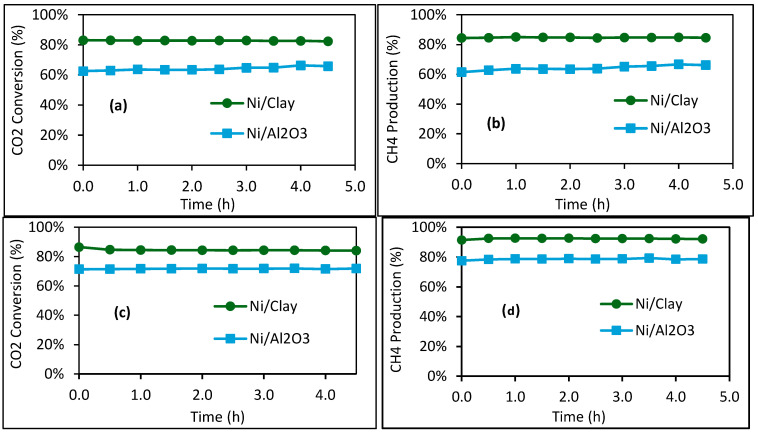
Carbon dioxide conversion and methane production at (**a**,**b**) 400 °C (for both in situ reduction and reaction temperature) and (**c**,**d**) 500 °C (for both in situ reduction and reaction temperature). Other conditions: 1 g of catalyst; inlet gas flowrate: 22.0 mL/min CO_2_, 90 mL/min H_2_, 37.5 mL/min N_2_, and 0 mL/min CH_4_; WHSV = 8970 mL·g_cat_^−1^·h^−1^.

**Figure 8 molecules-30-02110-f008:**
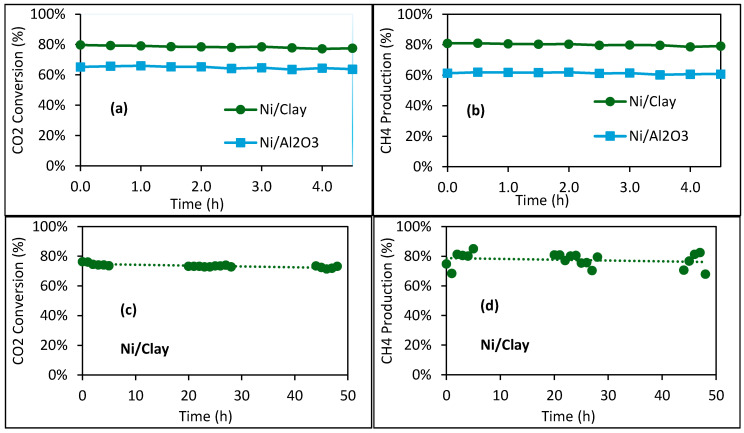
(**a**,**b**) Comparison of Ni/Clay and Ni/Al_2_O_3_ under the same conditions in the methanation reaction (in situ reduction at 500 °C; reaction temperature of 500 °C; 500 mg catalyst; inlet gas flowrate: 22.0 mL/min CO_2_, 90 mL/min H_2_, 37.5 mL/min N_2_, and 0 mL/min CH_4_). (**c**,**d**) Catalytic performance of Ni/Clay in the methanation reaction for a long reaction time (reaction conditions: in situ reduction at 500 °C; reaction temperature of 500 °C; 500 mg catalyst; inlet gas flowrate: 22.0 mL/min CO_2_, 90 mL/min H_2_, 3.75 mL/min N_2_, and 33.75 mL/min CH_4_; WHSV = 17,940 mL·g_cat_^−1^·h^−1^).

**Table 1 molecules-30-02110-t001:** XRF analysis of the natural clay; nd.: below detection limit.

Element	Na	Mg	Al	Si	P	S	K	Ca	Ti	Mn	Fe	Cu	Zn	Ba	Sr
Content (wt.%)	0.07	1.11	9.5	24.91	nd.	nd.	3.78	3.38	0.84	0.18	9.77	nd.	nd.	0.11	nd.

**Table 2 molecules-30-02110-t002:** List of applied catalytic tests and reaction conditions.

Entry	Catalyst	m_catalyst_ (g)/m_sintered Al2O3_ (g)	Total Catalyst Bed Height (cm)	In Situ Reduction Temperature, °C	Reaction Temperature, °C	Inlet Gas Mixture
#1	Ni/Clay	1.0/1.0	5.5	500	300	Mixture 1
#2	Ni/Al_2_O_3_	1.0/1.0	5.5	500	300	Mixture 1
#3	Ni/Clay_Cal	1.0/1.0	5.5	500	300	Mixture 1
#4	Ni/Al_2_O_3__Cal	1.0/1.0	5.5	500	300	Mixture 1
#5	Ni/Clay	1.0/1.9	5.5	400	400	Mixture 1
#6	Ni/Al_2_O_3_	1.0/1.9	5.5	400	400	Mixture 1
#7	Ni/Clay	1.0/1.9	5.5	500	500	Mixture 1
#8	Ni/Al_2_O_3_	1.0/1.9	5.5	500	500	Mixture 1
#9	Ni/Clay	0.5/1.0	2.75	500	500	Mixture 1
#10	Ni/Al_2_O_3_	0.5/1.0	2.75	500	500	Mixture 1
#11	Ni/Clay	0.5/1.0	2.75	500	500	Mixture 2

## Data Availability

Data are available on request.

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
