# Peer review of "Catalytic Methanation over Natural Clay-Supported Nickel Catalysts"

_molecules, 2025, doi:10.3390/molecules30102110_

Round 1

Reviewer 1 Report

Comments and Suggestions for Authors
  1. The paper does not clearly relate the limitations of existing industrial catalysts (such as Ni supported on Al₂O₃) to the advantages of natural clay. The innovative points are vague. The literature review is limited to traditional supports and should also focus on recent studies of natural clay-based catalysts, especially the impact of alkali metals on catalytic activity, which should be emphasized.
  2. From the TEM images, such as in Fig. 3e, it can be seen that the crystalline particles of Ni in the Ni/Al₂O₃ catalyst are larger than 20 nm, which is significantly larger than the Ni particles in the Ni/Clay catalyst. Combined with the SEM images, it is evident that after calcination, the NiO particles in Ni/Al₂O₃ are smaller, while in the reduced Ni/Clay, the particles of Ni are smaller. This could be the reason for the higher catalytic activity.
  3. Ni0 is the main active phase, and the XRD patterns of the catalysts after calcination and reduction should be provided to compare the changes.
  4. Hâ‚‚-TPR and COâ‚‚-TPD experiments should be provided to demonstrate the interaction between Ni and the support, as well as the COâ‚‚ adsorption capacity on the support.
  5. The selectivity change curve for methane in Fig. 7 should be added.
  6. It is recommended that the authors treat the clay with acid washing to remove the alkali metals and then load Ni to compare catalytic activity. This might better reflect the role of alkali metals. If possible, XPS characterization can be used to assist in proving the electronic interaction between the support and Ni.
  7. The paper should further supplement the characterization of the uncalcined catalyst after reduction, as this will help explain the reasons for the increased catalytic activity.
  8. The reference formatting is inconsistent. Please revise according to the journal's format.

Author Response

Manuscript ID: molecules-3544526

Answers to the questions of the reviewers (Round 1)

Reviewer 1:

  1. The paper does not clearly relate the limitations of existing industrial catalysts (such as Ni supported on Al₂O₃) to the advantages of natural clay. The innovative points are vague. The literature review is limited to traditional supports and should also focus on recent studies of natural clay-based catalysts, especially the impact of alkali metals on catalytic activity, which should be emphasized.

Answer: The literature review has been revised, and some paragraphs have been added to the revised manuscript to emphasize the motivation and importance of our research while attempting to keep the text clear and concise (see pages 2-3 of the revised manuscript).

  1. From the TEM images, such as in Fig. 3e, it can be seen that the crystalline particles of Ni in the Ni/Al₂O₃ catalyst are larger than 20 nm, which is significantly larger than the Ni particles in the Ni/Clay catalyst. Combined with the SEM images, it is evident that after calcination, the NiO particles in Ni/Al₂O₃ are smaller, while in the reduced Ni/Clay, the particles of Ni are smaller. This could be the reason for the higher catalytic activity.

Answer: We agree with this remark on the size of Ni nanoparticles, that has been mentioned in the manuscript.

  1. Ni0 is the main active phase, and the XRD patterns of the catalysts after calcination and reduction should be provided to compare the changes.

Answer: XRD of the reduced catalysts has been done and the results obtained have been added to the revised manuscript (pages 11-12 and Figures S4).

  1. Hâ‚‚-TPR and COâ‚‚-TPD experiments should be provided to demonstrate the interaction between Ni and the support, as well as the COâ‚‚ adsorption capacity on the support.

Answer: Supplementary Hâ‚‚-TPR and COâ‚‚-TPD experiments have been done and the results have been added to the revised manuscript (page 9-10 and Figure 3).

  1. The selectivity change curve for methane in Fig. 7 should be added.

Answer: According to this comment, the Eq. 3 has been revised and modified in order to take into account the presence of methane in the feeding Mixture 2 for the results in Fig. 8 in the revised manuscript.

  1. It is recommended that the authors treat the clay with acid washing to remove the alkali metals and then load Ni to compare catalytic activity. This might better reflect the role of alkali metals. If possible, XPS characterization can be used to assist in proving the electronic interaction between the support and Ni.

Answer: We agree with the reviewer on this suggestion. However, the treatment of clay with an acid solution to remove alkali metals seems to be a delicate operation. The beneficial impact of alkali metals in the methanation process has already been reported in the literature. So, we propose to do not do the supplementary experiment as suggested, but we have revised the interpretation of the results by providing more supporting data from the literature (see modification on page 15 of the revised manuscript).

  1. The paper should further supplement the characterization of the uncalcined catalyst after reduction, as this will help explain the reasons for the increased catalytic activity.

Answer: some more characterization results by ICP-AES, Hâ‚‚-TPR, COâ‚‚-TPD, XRD and XPS have been added to the revised manuscript to better supporting the explanation of the catalytic results.

  1. The reference formatting is inconsistent. Please revise according to the journal's format.

Answer: The references have been revised to meet the format required by the journal.

Reviewer 2 Report

Comments and Suggestions for Authors

This study proposes a novel approach for developing green catalysts by utilizing natural clay as an alternative to traditional alumina supports, combining its advantages of low cost and low environmental impact, while leveraging the potential promotional effects of natural mineral elements (e.g., Fe, K, Ca) in the clay. The non-calcined Ni/Clay catalyst exhibits 80% COâ‚‚ conversion and nearly 100% CHâ‚„ selectivity at 500°C, along with excellent 48-hour stability, outperforming the industrial alumina-supported catalyst (Ni/Alâ‚‚O₃), demonstrating practical application value.

1.The TEM results show a broad nickel particle size distribution (1-15 nm) in Ni/Clay. It is recommended to supplement statistical comparisons of particle sizes between calcined and non-calcined catalysts to clarify the effect of calcination on nickel aggregation.

2.The hypothesis that Fe, K, and other elements in the clay act as promoters lacks direct evidence (e.g., XPS or in-situ DRIFTS analysis). If feasible, selective leaching experiments or elemental doping control experiments should be conducted to validate the role of specific elements.

3.The long-term stability test did not address carbon deposition. Supplementary TGA or Raman spectroscopy analyses are advised to evaluate the catalyst’s resistance to coking.

4.While the superior activity of Ni/Clay at high temperatures (500°C) is attributed to the promotional effects of mineral elements in the clay, the authors did not explain why Ni/Alâ‚‚O₃ outperforms Ni/Clay at low temperatures (300°C). A discussion on the temperature-dependent activation of mineral elements should be added.

Author Response

Manuscript ID: molecules-3544526

Answers to the questions of the reviewers (Round 1)

Reviewer 2:

This study proposes a novel approach for developing green catalysts by utilizing natural clay as an alternative to traditional alumina supports, combining its advantages of low cost and low environmental impact, while leveraging the potential promotional effects of natural mineral elements (e.g., Fe, K, Ca) in the clay. The non-calcined Ni/Clay catalyst exhibits 80% COâ‚‚ conversion and nearly 100% CHâ‚„ selectivity at 500°C, along with excellent 48-hour stability, outperforming the industrial alumina-supported catalyst (Ni/Alâ‚‚O₃), demonstrating practical application value.

1.The TEM results show a broad nickel particle size distribution (1-15 nm) in Ni/Clay. It is recommended to supplement statistical comparisons of particle sizes between calcined and non-calcined catalysts to clarify the effect of calcination on nickel aggregation.

Answer: We agree with this relevant remark. However, we could not perform further TEM analysis by logistic reason of these measurements (unavailability of the apparatus for reduced nickel catalysts). On the other hand, we have compared the crystallite size of nickel formed by calcination under the air and by direct reduction of the nickel precursor under H2, using XRD analysis. The results have been added to the revised manuscript. The calcination step only slightly increased the average size of nickel crystallites.

2.The hypothesis that Fe, K, and other elements in the clay act as promoters lacks direct evidence (e.g., XPS or in-situ DRIFTS analysis). If feasible, selective leaching experiments or elemental doping control experiments should be conducted to validate the role of specific elements.

Answer: We agree that selective leaching experiments or elemental doping control experiments can help to better highlight the impact of the metals initially present in the clay matrix. However, leaching tests are delicate and can impact the structure of the clay matrix, while the impact of Fe, K and other elements on the activity of nickel-based catalysts have already been reported in the literature and have mentioned and cited in the manuscript. About in-situ DRIFTS, unfortunately, we have not the possibility to the analysis. On the other hand, XPS analysis has been done and the results have been added to the revised manuscript. Moreover, supplementary SEM-EDX analyses have been done to highlight the presence of these mineral elements on the surface of the clay support.

3.The long-term stability test did not address carbon deposition. Supplementary TGA or Raman spectroscopy analyses are advised to evaluate the catalyst’s resistance to coking.

Answer: Supplementary TGA results have been added to the revised manuscript to highlight the coke resistance of the Ni/Clay catalyst. This result can be found in the Figure S7 (a) of the Supplementary Materials.

  1. While the superior activity of Ni/Clay at high temperatures (500°C) is attributed to the promotional effects of mineral elements in the clay, the authors did not explain why Ni/Alâ‚‚O₃ outperforms Ni/Clay at low temperatures (300°C). A discussion on the temperature-dependent activation of mineral elements should be added.

Answer: Mineral elements, which are present in the clay support (and absent in alumina support), might be inactivated at 300 °C. For example, RWGS reaction is generally carried out over iron-based catalyst around 600 °C. So, the fact that Ni/Alâ‚‚O₃ outperforms Ni/Clay at 300 °C was attributed to the different on the average size of nickel particle. This has been commented in the present manuscript.

Reviewer 3 Report

Comments and Suggestions for Authors

The manuscript studies the use of a natural clay as support for Ni catalysts for CO2 methanation reaction. The solid is compared to one similar prepared on a commercial alumina support.

Differences in catalytic behaviour could be related to differences in Ni loading, Ni dispersion (particle size), Ni-support interaction and presence of other metals (in the case of the natural clay)

The paper could be publishable if the authors improve it clarifying some of these points

A.- Real Ni loadings must be measured

B- Ni dispersion (Ni particle size) of all reduced samples must be determined

C- Changes in Ni size must be clearly related to the characteristics of both supports(BET, porosity, presence of metal,etc.)

Some specific points

1.- Has been the natural clay submitted to any treatment of cleanness or separation of the impurities? Or it has been used as extracted from nature?

Which is its variability in composition? Is a representative sample?

2.- Why the stability test is the only using the Mixture 2

3.- line 41.  Rb is not a noble metal, but an alkaline one

4.- line 116. 600ºC is not evalauted in this paper

5.- Reaction products are not evaluated on-line, but after collection in a tedlar gas bag.

Which volume? Are really representative?

Have the authors used an internal standard?

In any case, Carbon balance must be commented.

6.- Catalysts powder is embedded in two layers of alumina. So, sample is not diluted.

Then, alumina is not acting as diluyent, and it is not participating as an important subject in the catalytic process.

a Which is it role?

b) Diffusional problems could occur. Have the authors evaluated them?

  1. Figure 1 is not necessary. It can be removed

8.- Figure 3. TEM.

  1. a) The Ni images and Ni particle distribution of reduced Ni/Clay_cal and Ni/Al2O3_cal must also be shown and discussed. Authors said in lines 227-230 that differences must exist, so TEM is mandatory to be done.
  2. b) In the microgrpahs, Ni particles are not clearly shown. Plase, indicate them

9.-XRD of reduced samples must be shown and discussed.

10.- Ni and NiO crystallite sizes must be calculated from Scherrer equation  and compared to TEM values

Author Response

Manuscript ID: molecules-3544526

Answers to the questions of the reviewers (Round 1)

Reviewer 3:

The manuscript studies the use of a natural clay as support for Ni catalysts for CO2 methanation reaction. The solid is compared to one similar prepared on a commercial alumina support.

Differences in catalytic behaviour could be related to differences in Ni loading, Ni dispersion (particle size), Ni-support interaction and presence of other metals (in the case of the natural clay)

The paper could be publishable if the authors improve it clarifying some of these points

A.- Real Ni loadings must be measured

Answer: Nickel content of the catalysts have been added to the revised manuscript (page 7).

B- Ni dispersion (Ni particle size) of all reduced samples must be determined

Answer: We agree with this relevant remark. However, we could not perform further TEM analysis by logistic reason of these measurements. On the other hand, we have compared the crystallite size of nickel formed by calcination under the air and by direct reduction of the nickel precursor under H2. The results have been added to the revised manuscript. The calcination step only slightly impacts the size of nickel crystallites.

C- Changes in Ni size must be clearly related to the characteristics of both supports (BET, porosity, presence of metal, etc.)

Answer: The difference in Ni particle size may principally be due to the difference in the textural properties of the two supports. Alumina with high specific surface area (120 m2/g) favors better the dispersion of nickel nanoparticles than clay (20 m2/g). This has been added to the revised manuscript.

Some specific points

Answer:

1.- Has been the natural clay submitted to any treatment of cleanness or separation of the impurities? Or it has been used as extracted from nature?

Answer: After the extraction, clay sample was dried at 105 °C overnight, then, ground and sieved to have a maximum particle size of 500 µm. This fraction was used to prepare the clay-supported nickel catalysts.

Which is its variability in composition? Is a representative sample?

Answer: This was not checked, but we have extracted a large sample of some kg for this study.

2.- Why the stability test is the only using the Mixture 2

Answer: The main objective is to find the best catalyst and the best operation conditions to do the methanation of the simulated biogas (which mainly contains CO2 and CH4). However, CH4 is also the main targeted product of the methanation, so for the parametric study, only the Mixture 1 without CH4 was used to easily compare the catalysts. For the stability test, the best catalyst and the best conditions have been validated with the simulated biogas (Mixture 2).

3.- line 41.  Rb is not a noble metal, but an alkaline one

Answer: This has been corrected in the manuscript (Ru but not Rb).

4.- line 116. 600ºC is not evaluated in this paper

Answer: Effectively we did not do the tests at 600 °C. The temperature range of the catalytic tests has been corrected in the revised manuscript.

5.- Reaction products are not evaluated on-line, but after collection in a tedlar gas bag.

Which volume? Are really representative?

Answer: It is a standard procedure for offline analysis by µ-GC. Tedlar bag is a standard device to correctly collect the gas mixture from the reactor outlet. Before collection, the bag is purged three time with nitrogen and emptied also three time using a vacuum pump. Various bag sizes exist and, in this study, the bag of 200 mL volume was used. Part of the gas in the bag was injected to the µ-GC for the analysis.

Have the authors used an internal standard?

Answer: Yes, nitrogen has been used as internal standard for µ-GC analysis. This has been specified in the revised manuscript.

In any case, Carbon balance must be commented.

Answer: Carbon balance has been added to the manuscript for the validation test (Figure S7 (b) in the Supplementary materials).

6.- Catalysts powder is embedded in two layers of alumina. So, sample is not diluted.

Answer: No, there was any dilution.

Then, alumina is not acting as diluent, and it is not participating as an important subject in the catalytic process.

a Which is it role?

Answer: Embedding the catalyst in two layers of alumina powder is only to secure the stability of the catalyst bed (i.e. to keep the bed fixed at the center of the reactor tube).

  1. b) Diffusional problems could occur. Have the authors evaluated them?

Answer: Catalyst particles smaller than 500 µm were used. This dimension guarantees the ratio of inner diameter of the reactor to grain diameter superior to 10, which is conventionally applied in the literature. No diffusional test was performed.

Figure 1 is not necessary. It can be removed

Answer: Figure 1 has been moved to the Supplementary Materials.

8.- Figure 3. TEM.

  1. a) The Ni images and Ni particle distribution of reduced Ni/Clay_cal and Ni/Al2O3_cal must also be shown and discussed. Authors said in lines 227-230 that differences must exist, so TEM is mandatory to be done.

Answer: We agree with this relevant remark. However, we could not perform further TEM analysis by logistic reason of these measurements (unavailability of the equipment for reduced nickel catalysts). On the other hand, we have compared the crystallite size of nickel formed by calcination under the air and by direct reduction of the nickel precursor under H2. The results have been added to the revised manuscript. The calcination step only slightly increased the size of nickel crystallites.

  1. b) In the micrographs, Ni particles are not clearly shown. Please, indicate them

Answer: Red arrows have been added to indicate the position of these particles.

9.-XRD of reduced samples must be shown and discussed.

Answer: XRD results of the reduced catalysts have been added to the revised manuscript (pages 11-12 and Figure S4).

10.- Ni and NiO crystallite sizes must be calculated from Scherrer equation and compared to TEM values

Answer: Ni crystallite sizes have been calculated from Scherrer equation and the results have been added to the revised manuscript (pages 11-12).

Round 2

Reviewer 1 Report

Comments and Suggestions for Authors

The authors have revised this manuscript based on my comments. Thus I recommend it to be accepted by Molecules.

Reviewer 3 Report

Comments and Suggestions for Authors

The authors have reasonably improved the article according to the referee's suggestions.